# Study on Laser Surface Hardening Behavior of 42CrMo Press Brake Die

**Huizhen Wang \*, Yuewen Zhai, Leyu Zhou and Zibo Zhang**

Beijing Research Institute of Mechanical and Electrical Technology, Beijing 100083, China; zhaiyw@jds.ac.cn (Y.Z.); zhouleyu@ustb.edu.cn (L.Z.); zhangzb@brimet.ac.cn (Z.Z.)

\* Correspondence: wanghzh@jds.ac.cn; Tel.: +86-10-82415069

**Abstract:** Laser surface hardening is a promising surface technology to enhance the properties of surfaces. This technology was used on the 42CrMo press brake die. Its hardening behavior was investigated by using scanning electron microscopy and electron backscattering diffraction. The results indicated that the martensite in the hardening zone was significantly finer than that in the substrate. There were many low-angle grain boundaries in the martensite of the hardening zone, and the kernel average misorientation and grain orientation spread in the hardening zone grains were obviously greater, which further improved the hardness of the hardening zone, especially near the substrate. The microstructure and the properties of the blade maintained excellent uniformity with treatment by single-pass laser surface hardening with a spot size of 2 mm, scanning speed of 1800 mm/min, and power of 2200 W. The hardness of the hardening zone was 1.6 times higher than that of the base material, and the thickness of the hardening zone reached 1.05 mm.

**Keywords:** press brake die; laser surface hardening; hardening zone; microstructure; martensite





## 1. Introduction

The 42CrMo steel is widely used as press brake dies due to its excellent performance of high strength and toughness. Its usual process is conventional quenching and tempering. However, with the increasing requirements of properties under some working conditions, the blade of the press brake die requires a higher wear resistance with minimal deformation. Therefore, improving the surface properties is of great significance, while conventional quenching and tempering come with some limitations [1].

Increasing the hardness is an effective way to improve wear resistance. There have been several surface modification technologies used to enhance the surface properties of metallic materials, such as carburizing, nitriding, carbo-nitriding, induction hardening, and laser surface treatment [2–5]. One of the promising surface technologies is laser surface hardening, which has advantages of high accuracy, high thermal concentration, low heat input, and high processing speed [6,7]. Importantly, it can significantly enhance the surface hardness and wear resistance [8–12]. This technology has been successfully applied in many kinds of materials to improve their properties. Khorram and Junaid et al. [13,14] investigated the laser surface processes of Ti-5Al-2.5Sn alloy. They reported that the presence of the martensitic phase in fusion and heat-effected zones caused an increase in the hardness compared to the base metal. Moradi and Mahmoudi et al. [15,16] worked on the laser surface hardening of AISI 420 martensitic stainless steel and concluded that this process can increase the hardness and corrosion resistance of this steel. Levcovici et al. [17] reported that the hardness of austenitic stainless steel with laser surface hardening was 2.5 to 3 times higher than that of the base material. Lee and Telasang et al. [18,19] studied the laser surface hardening behavior of AISI H13 tool steel and the results showed that the hardness and the wear resistance were enhanced. Li and Lusquinosa et al. [2,20] investigated the properties of AISI 1045 steel after the laser surface process. According to their work, the surface hardness was improved significantly and the process using a

diode laser would result in a higher quality than the $CO_2$ one. Chen et al. [21] studied the behaviors of 40Cr steel by laser quenching on impact abrasive wear and the results showed that following laser quenching, the hardness of the quenched region was significantly increased, which resulted in the improvement of impact abrasive wear resistance.

However, less research has been performed on the laser surface quenching of 42CrMo, especially for the blade of 42CrMo press brake die. Due to its small radius, it is difficult to obtain enough thickness of the hardened layer and uniform distribution on both sides of the blade. Meanwhile, the impact mechanism of the microstructure on surface properties has seldom been investigated. In this paper, the mechanical properties of 42CrMo press brake die after laser surface hardening were studied. In addition, the electric back-scatter diffraction (EBSD) testing technology was used to explore in depth the influence of the microstructure on the surface hardness and the characteristics of surface martensite transformation. This study has significance for the applications of 42CrMo press brake die and the laser surface hardening process formulation and evaluation of many steels.

## 2. Materials and Methods

The material used in this study was 42CrMo steel. Its chemical composition based on GB/T 3077-2015 is listed in Table 1. The press brake die is as shown in Figure 1a, and Figure 1b shows the local enlarged drawing of the blue box in Figure 1a. The press brake die was quenched at 1133 K and tempered at 553 K. The hardness was 45HRC. Then, the blade was processed by laser surface hardening.

**Table 1.** Chemical composition of 42CrMo steel (wt.%).

| C | Si | Mn | P | S | Cr | Mo | Fe |
|---|---|---|---|---|---|---|---|
| 0.38~0.45 | 0.17~0.37 | 0.5~0.8 | <0.035 | <0.04 | 0.90~1.20 | 0.15~0.25 | Balance |

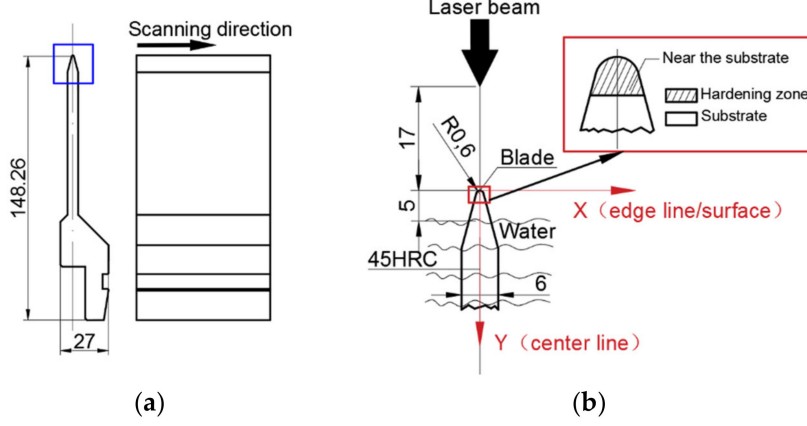

**Figure 1.** (**a**) General diagram and (**b**) local enlarged drawing of press brake die (unit of measurement: mm).

Laser surface hardening was carried out on a Trudiode 4006 diode laser (Trumpf, Ditchingen, Germany). According to the research of Li et al. [1], the thickness of the hardened layer is larger when the initial microstructure is martensite. In this paper, the initial microstructure was tempered martensite, which contributed to obtain a thick hardened layer. Based on previous research, the 42CrMo processed with a spot size of 2 mm, a scanning speed of 1800 mm/min, and a laser power of 2200 W showed excellent performance, and these process parameters were used in this paper. In addition, Lin et al. [22] reported that the water cooling during the laser surface process can furthermore enhance the wear resistance due to the accumulative contribution of grain refinement and dislocation strengthening. Therefore, auxiliary water cooling was used in the experiment,

as shown in Figure 1b. The blade was treated by a single-pass laser along the scanning direction of Figure 1a.

The specimens were ground, polished in 5% $HClO_4$, and etched in 4% Nital solution. An optical microscope (Zeiss Axio Scope. A1, Carl Zeiss AG, Jena, Germany) and a scanning electron microscope (Zeiss GeminiSEM 500 field emission microscopy, Carl Zeiss AG, Jena, Germany) equipped with HKL EBSD systems were used to evaluate the microstructures. The micro hardness was tested on the INNOVATEST 423D micro-Vickers hardness tester (INNOVATEST Europe BV, Maastricht, The Netherlands). The observation position coordinates, which were described as (X, Y), are based on the X-Y coordinate system in Figure 1b and have been marked in the figures.

### 3. Results

#### 3.1. Mechanical Properties of Laser Surface Hardening Zone

The hardness distribution along the center line of Figure 1b shows that the hardness of the hardening layer reached 695–734 HV0.2, which is 1.6 times higher than that of the base material (Figure 2a). The thickness of the hardening zone reached 1.05 mm. This implies that the laser hardening was obvious. In addition, the hardness near the substrate was slightly higher than that near the surface. The uniformity distribution of the hardness is shown in Figure 2b. The hardness at 0.6 mm from the surface was $713 \pm 4$ HV0.2, which demonstrates good uniformity.

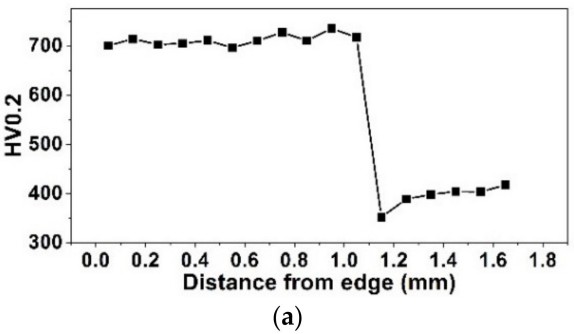
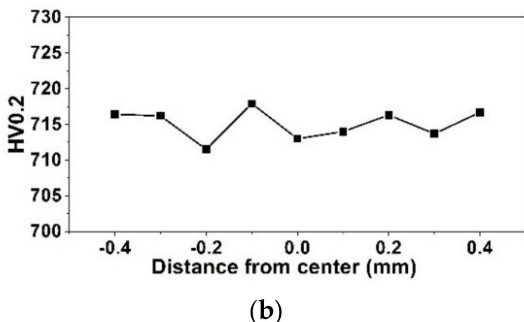

(**a**)            (**b**)

**Figure 2.** Hardness distribution of blade (**a**) along the center line and (**b**) at 0.6 mm from the surface.

For the blade of press brake die, it is important to select the appropriate spot size, scanning power, and scanning speed. Due to the blade with an arc shape and small radius, tempering is produced easily. In order to avoid this, one must choose the appropriate spot size, and one-pass scanning is the best way of laser surface quenching. At the same time, this method can solve the problem of uneven performance on both sides.

#### 3.2. Microstructure Evolution of Laser Surface Hardening Zone

##### 3.2.1. Microstructure Evolution from Hardening Zone to Substrate

The microstructure of the blade from the surface to the substrate is shown in Figure 3. SEM and EBSD detectors were used to detect the microstructure and the orientation at various positions in the X-Y coordinate system of Figure 1b, and the position coordinates have been marked in the microstructure maps. According to the microstructure transition from the surface to the substrate, two kinds of martensite existed in the hardening zone (Figure 3). The martensite at 0.1 mm from the surface in the hardened zone was the lath one; the lath martensite at 0.6 and 0.8 mm was smaller than that at 0.1 mm; meanwhile, there was a high amount of near-equiaxed martensite. Figure 4 shows that the aspect ratio of the martensite lath at the 5 mm substrate was larger than that at the hardening zone, which means the martensite lath at the 5 mm substrate was slenderer. As shown in Figure 5, the grain boundaries of the original austenite were obtained according to the Kurdjumov–Sachs orientation relationship. The original austenite at the 0.1, 0.6, and 0.8 mm hardening zone was obviously finer than that at 5 mm, especially at 0.6

and 0.8 mm. Thus, the microstructure of the hardening zone was characterized by two kinds of features: the coarse martensitic lath near the surface and the mixed microstructure of lath martensite and near-equiaxed martensite at the hardening zone near the substrate. The reasons for the formation of the two types of martensite are as follows.

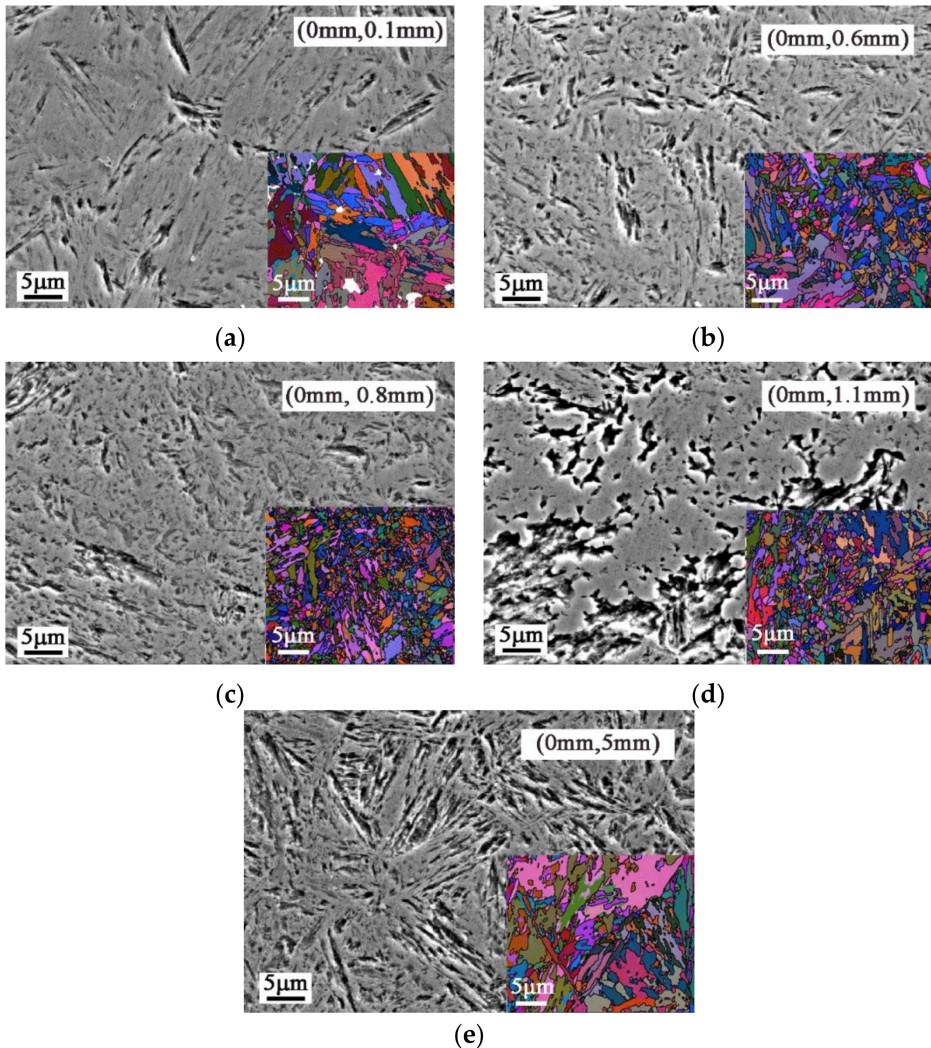

**Figure 3.** Microstructure at (**a**) 0.1 mm, (**b**) 0.6 mm, (**c**) 0.8 mm, (**d**) 1.1 mm, and (**e**) 5 mm from the surface.

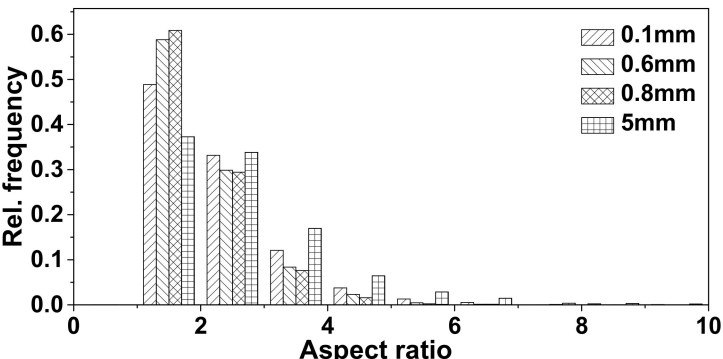

**Figure 4.** Aspect ratio distribution of martensite at 0.1, 0.6, 0.8, and 5 mm from the surface.

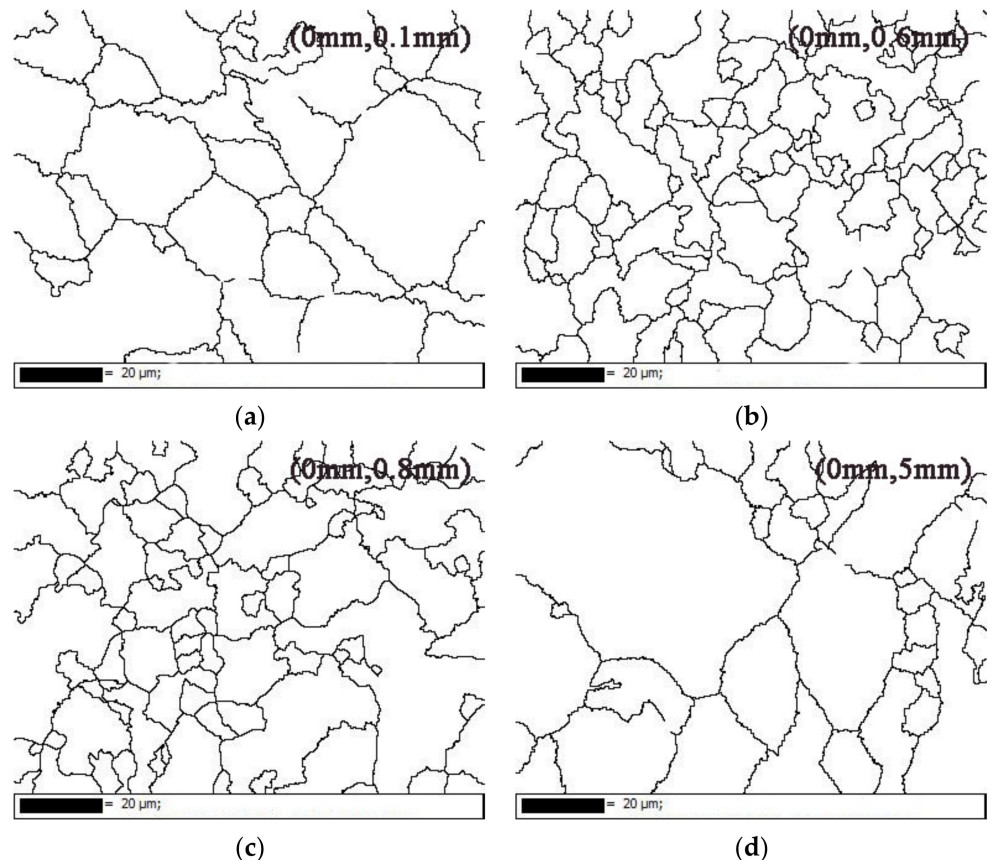

**Figure 5.** Grain boundary map of primary austenite at (**a**) 0.1 mm, (**b**) 0.6 mm, (**c**) 0.8 mm, and (**d**) 5 mm from the surface.

1. The surface temperature of the hardening zone increased rapidly due to the short-time action of the high-energy laser, and austenite grains generated quickly. As a result of the high-speed quenching induced by the cold substrate, fine austenite was formed, which subsequently transformed into fine martensite.
2. The growth of the original austenite near the surface was more sufficient than that near the substrate, while the heat conduction near the substrate was faster, so finer martensite formed near the substrate. This is clearly reflected in Figure 3 by EBSD maps.

Therefore, due to the finer martensite, the hardness of the hardening zone was significantly higher than that of the substrate, and the hardness near the substrate was also slightly higher than that near the surface.

Based on the EBSD detection, the difference between the martensite in the hardening zone and in the substrate was analyzed. Figures 6 and 7 show the distribution of martensite grain boundaries from the surface to the substrate. Compared to the substrate, there were more low-angle grain boundaries in the hardening zone, especially the 2°–5° grain boundaries marked by the red line (Figure 6). The small-angle grain boundaries contributed to the improvement of hardness at the surface zone, especially at 0.6 and 0.8 mm.

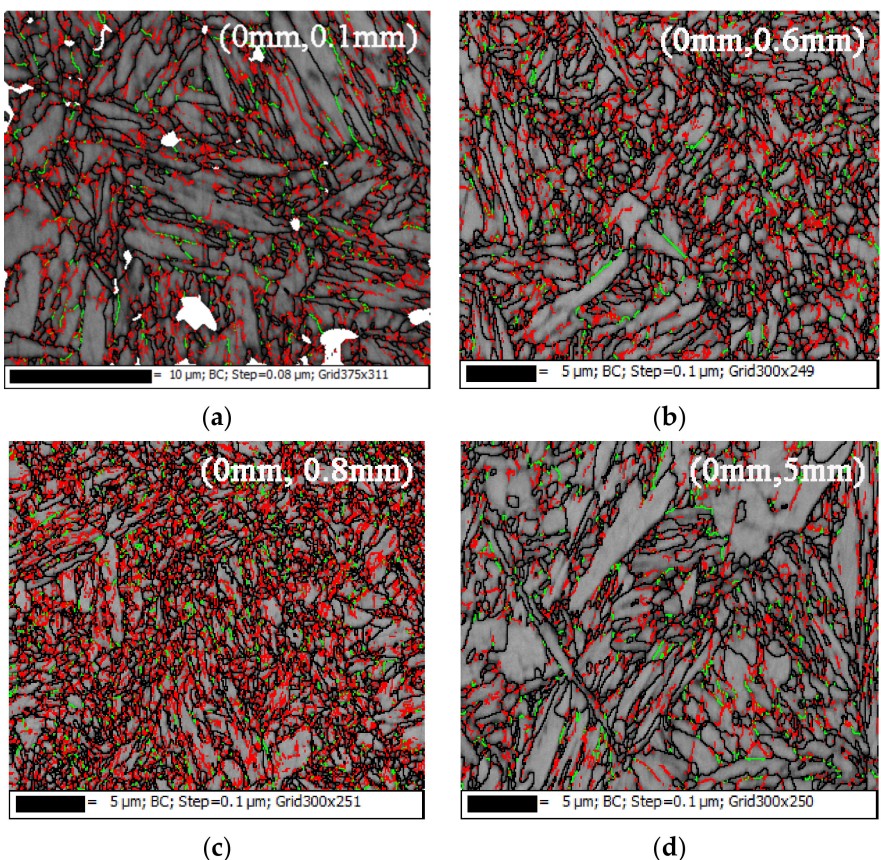

**Figure 6.** Grain boundary map of martensite at (**a**) 0.1 mm, (**b**) 0.6 mm, (**c**) 0.8 mm, and (**d**) 5 mm from the surface (red lines: 2°–5°, green lines: 5°–10°, black lines: >10°).

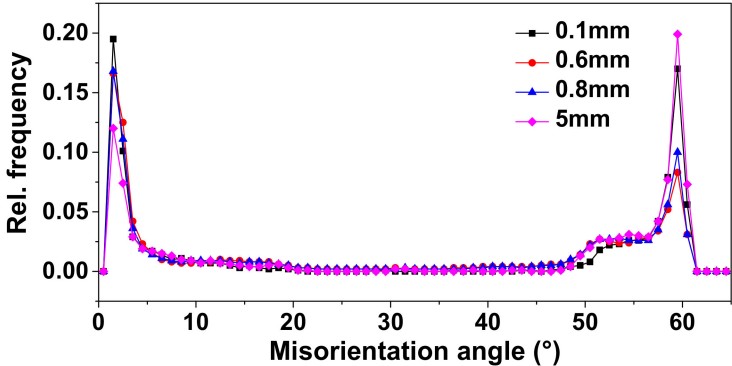

**Figure 7.** Misorientation distribution of martensite at 0.1, 0.6, 0.8, and 5 mm from the surface.

The kernel average misorientation (KAM) distributions in Figures 8 and 9 reflect the effect of the stress distribution in the hardening zone and the substrate on the properties. From the surface to near the substrate, the kernel average misorientation increased gradually. The KAMs with the highest relative frequency (rel. frequency) were 0.66° at 0.1 mm, 0.76° at 0.6 mm, and 1.04° at 0.8 mm, which was larger than 0.57° at 5 mm. The surface of the specimen was rapidly heated to the austenitizing temperature by a laser, and it was quenched to generate fine martensite by a cold substrate and auxiliary water-cooling. Especially, close to the substrate position, the quenching effect was more significant, the martensite was also finer, and the internal stress was higher, so the hardness property of the hardening zone was higher than that of the substrate, especially near the substrate.

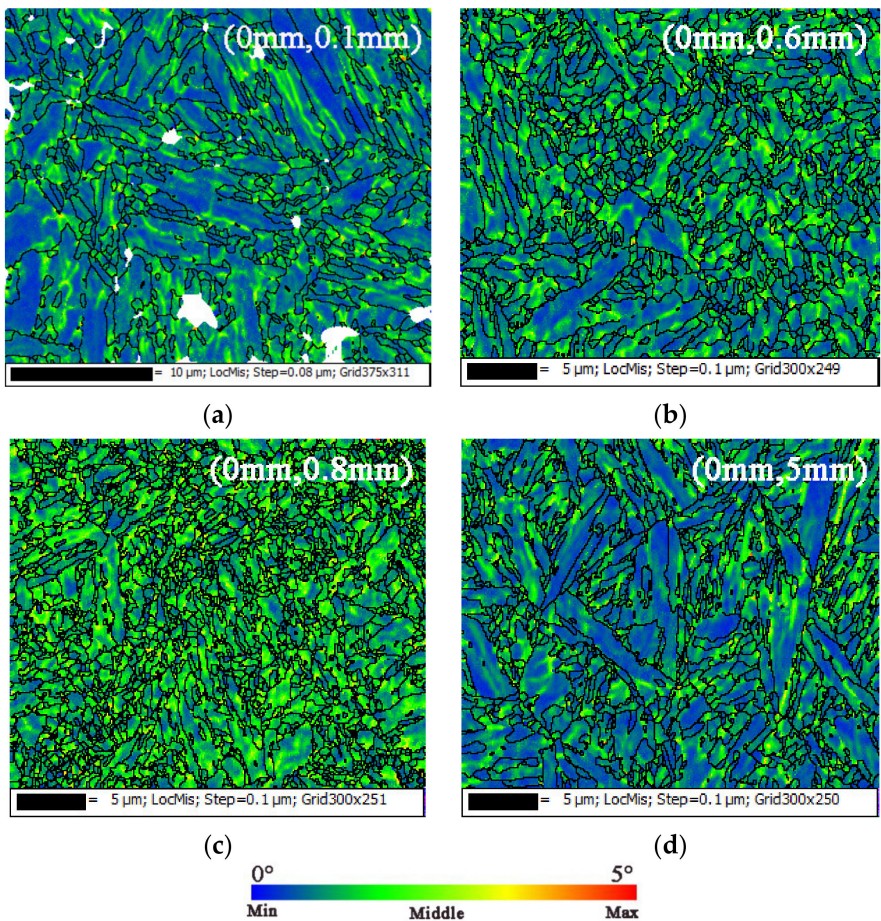

**Figure 8.** KAM map of martensite at (**a**) 0.1 mm, (**b**) 0.6 mm, (**c**) 0.8 mm, and (**d**) 5 mm from the surface (black lines: >10°).

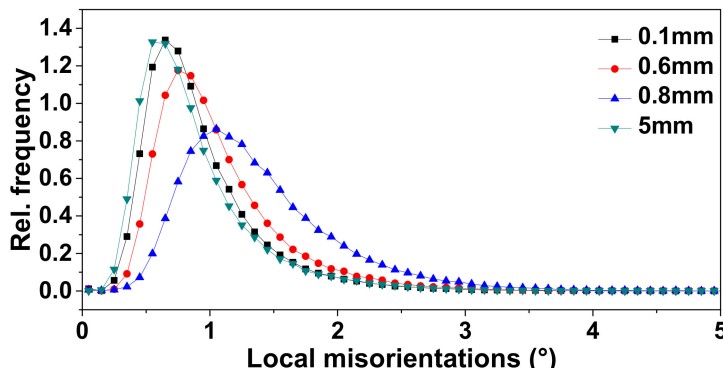

**Figure 9.** KAM distribution of martensite at 0.1, 0.6, 0.8, and 5 mm from the surface.

Figure 10 shows the grain orientation spread (GOS) in martensite, which is the average value of the misorientation between each point orientation and the average orientation in the grains, which reflects the internal distortion and stress of the grains. The GOS was mainly distributed in the range of 0.6°–1.3° in the hardening zone and 0.3°–0.6° in the 5 mm substrate, which implies that the internal distortion and stress in the grains of the hardening zone were significantly greater than those in the grains of the substrate. This indicates that the grains of the hardening zone had higher hardness properties.

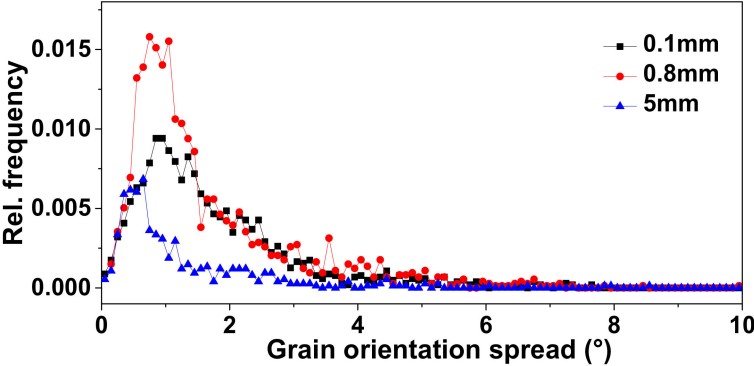

**Figure 10.** GOS distribution of martensite at 0.1, 0.6, 0.8, and 5 mm from the surface.

In conclusion, due to the instantaneous action of the high-energy laser, the surface temperature of the specimen rapidly increased to the austenitizing temperature. Combined with the self-quenching substrate and auxiliary water cooling, finer original austenite grains of the hardening zone were generated, and a mixed microstructure of fine lath martensite and near-equiaxed martensite formed near the substrate. In addition to the grain size advantage, there were more low-angle grain boundaries in martensite of the hardening zone, and the KAM and GOS in the grain were significantly greater than the substrate grain, which implies that there were more dislocations, distortion, and internal stress in martensite of the hardening zone, especially near the substrate. Thus, the internal microstructure characteristics of martensite in the hardening zone further improved the hardness of the hardening zone.

### 3.2.2. Uniformity Distribution of Microstructure and Properties of Hardening Zone

Figures 11 and 12 show the characteristics of the microstructure of the bending die blade at 0.6 mm from the surface. The microstructure distribution along the *X*-axis at 0.6 mm from the surface was both fine lath martensite and near-equiaxed martensite (Figure 11). On the other hand, the low-angle grain boundaries were mainly concentrated at about 1.5°, the KAM was concentrated at 0.75°–0.86°, and the GOS within the grain was distributed at about 0.74°. This implied that the internal microstructure of the grain also had satisfactory uniformity. Thus, a single-pass laser surface hardening of the press brake die blade with a spot size of 2 mm, scanning speed of 1800 mm/min, and power of 2200 W can solve the problems of insufficient process hardness and uneven distribution of properties and microstructure, and according to different requirements of laser hardening depth, the spot size and the process parameters can be optimized.

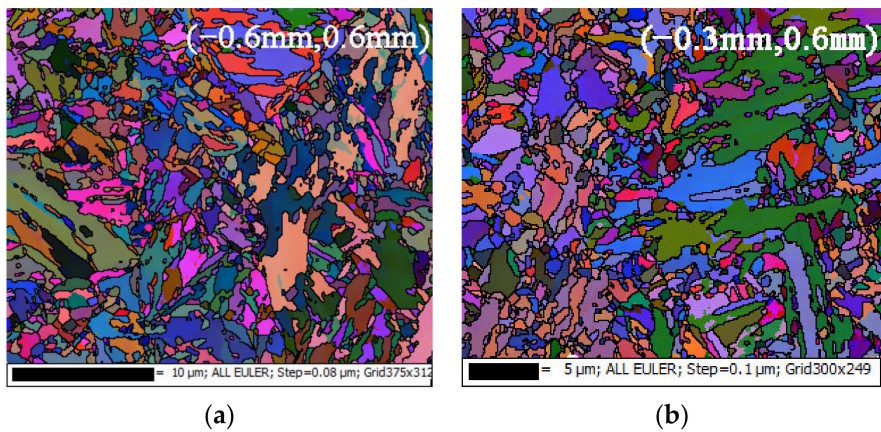

(a)　　　　　　　　　　　　　　　　　　　　　　　　　　(b)

**Figure 11.** *Cont.*

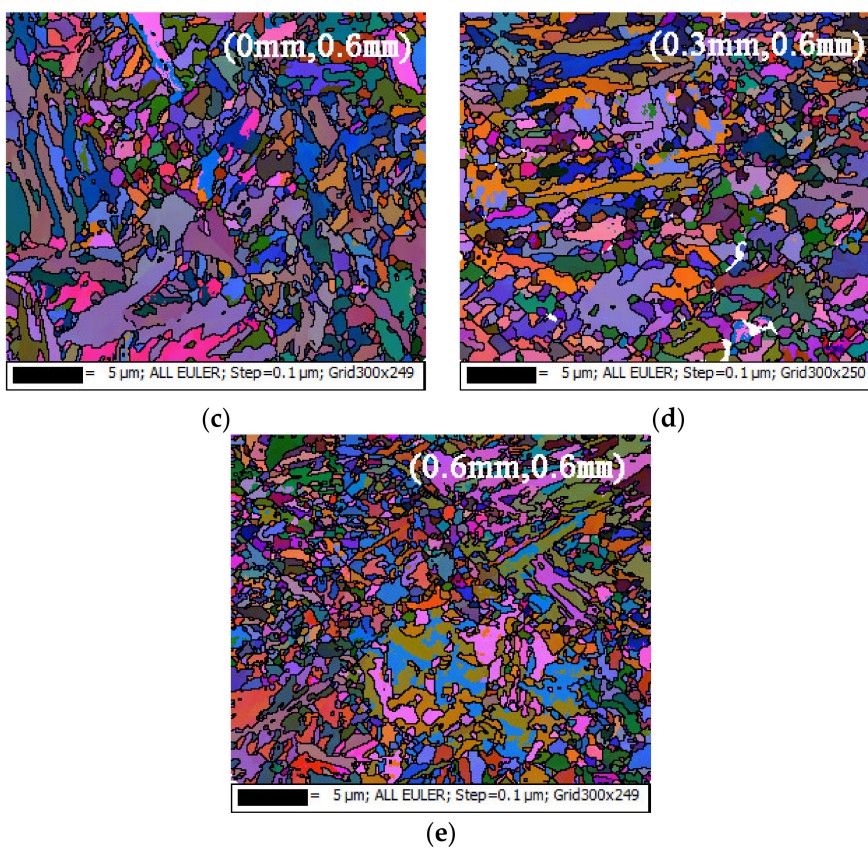

**Figure 11.** Uniformity distribution of microstructure at (**a**) (−0.6 mm, 0.6 mm), (**b**) (−0.3 mm, 0.6 mm), (**c**) (0 mm, 0.6 mm), (**d**) (0.3 mm, 0.6 mm), and (**e**) (0.6 mm, 0.6 mm).

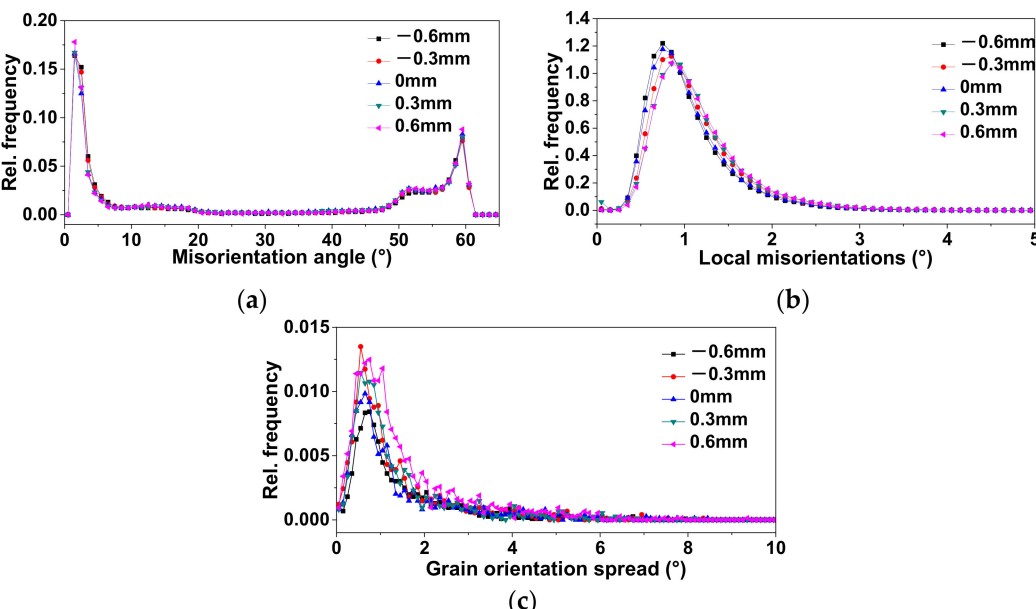

**Figure 12.** Distribution of (**a**) misorientation, (**b**) KAM, and (**c**) GOS at different positions from the surface of 0.6 mm.

## 4. Conclusions

The blade of the 42CrMo press brake die was hardened by the diode laser. The characteristics of the properties and the microstructure from the hardening zone to the substrate were analyzed. The main results are as follows:

(1) The blade of 42CrMo press brake die with appropriate process parameters of laser surface hardening can obtain excellent microstructure and properties. The hardness of the hardening zone was 1.6 times higher than that of the base material, and the thickness of the hardening zone reached 1.05 mm. The hardness and the microstructure distribution were uniform. Laser surface hardening with optimized process parameters solved the problem of the insufficient and uneven distribution of properties of the 42CrMo press brake die blade.

(2) The martensite in the hardening zone was remarkably finer than that in the substrate. Due to the faster heat conduction, ultrafine martensite formed near the substrate. Meanwhile, there were many low-angle grain boundaries in martensite of the hardening zone, and the KAM and GOS in the grains were obviously greater than those in the substrate grains, especially near the substrate. This implies that there were more dislocations, distortion, and internal stress in martensite of the hardening zone, which further improved the hardness of the hardening zone.

**Author Contributions:** Conceptualization, H.W.; methodology, H.W. and L.Z.; software, H.W.; validation, H.W.; formal analysis, H.W.; investigation, H.W.; resources, Y.Z.; data curation, H.W. and Z.Z.; writing—original draft preparation, H.W.; writing—review and editing, L.Z. and Y.Z.; visualization, H.W.; supervision, L.Z. and Y.Z.; project administration, L.Z. and Y.Z.; funding acquisition, L.Z. and Y.Z. All authors have read and agreed to the published version of the manuscript.

**Funding:** This research was funded by the National Key Research and Development Program of China, Grant No. 2020YFB2010300.

**Institutional Review Board Statement:** Not applicable.

**Informed Consent Statement:** Not applicable.

**Data Availability Statement:** Data presented are original.

**Conflicts of Interest:** The authors declare no conflict of interest.

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
