# Peer review of "Study on Laser Surface Hardening Behavior of 42CrMo Press Brake Die"

_coatings, doi:10.3390/coatings11080997_

Round 1
Reviewer 1 Report
I suggest to the authors to improve the introduction, which is too synthetic, does not provide a frame in which the authors’ work is inserted, and does not fully describe the aim of the work and its novelty.
- The Fig. 1 should be remade with better quality (higher dpi) and better presentment what was in this picture. This also applies to the Fig. 2.
- Overall all pictures are blurry, it is hard to see anything on them. I suggest improving them much since its hard to see the axial values and further judge about its correctness.
- The discussion part could be expanded with in-deep enunciation about the results.
Author Response
Dear reviewers,
Thank you for your comments. Those comments are valuable and very helpful. We have read through comments carefully and have made corrections.
We would love to thank you for allowing us to resubmit a revised copy of the manuscript and we highly appreciate your time and consideration.
Sincerely.
Huizhen Wang.
Response to Reviewer 1 Comments
Point 1: I suggest to the authors to improve the introduction, which is too synthetic, does not provide a frame in which the authors’ work is inserted, and does not fully describe the aim of the work and its novelty.
Response 1: The introduction was revised. In the introduction, the important work was added. And the aim is clearer. Please see the introduction for details.
Point 2: The Fig. 1 should be remade with better quality (higher dpi) and better presentment what was in this picture. This also applies to the Fig. 2.
Response 2: Fig. 1 was drawn again, the relevant information of Fig. 2 and Fig. 3 was combined in Fig. 1, and Fig. 2 and Fig. 3 were deleted. Please see the article for details.
Point 3: Overall all pictures are blurry, it is hard to see anything on them. I suggest improving them much since its hard to see the axial values and further judge about its correctness.
Response 3: I modified the pictures to make them clearer. Please see the article for details.
Point 4: The discussion part could be expanded with in-deep enunciation about the results.
Response 4: Some revisions have been made in the discussion section. Please see the article for details. In the subsequent work, I will study the distribution of stress field and temperature field through numerical simulation, and combine it with microstructure and properties to further analyse the behaviour of laser surface hardening.

Reviewer 2 Report
Notes to the authors:
Table 1.
Please indicate the standard according to which the chemical composition of the tested material was given, as well as specify the heat treatment after which the material was subjected to laser hardening.
Fig. 1.
The part subjected to laser treatment should be presented in accordance with the technical drawing - projections, dimensions. It is also necessary to mark on the drawing the laser processed surfaces, the trajectory of the laser beam, and the angle of incidence of the beam (it is known that the laser beam is directed perpendicular to the plane being processed.) In this case, when the edge of the element is rounded, it is quite a complicated machining procedure. If the orientation of the laser beam relative to the blade is perpendicular only to the tangent plane to the blade, you will never obtain a uniformly heated layer on the side surfaces of the blade and thus evenly hardened layers (positioning of the laser beam relative to the blade is shown in Fig. 3), as mentioned in article.
The figure should also show e.g. 1a. what the surface layer looks like after conventional hardening and tempering (its dimensions and microhardness), and then compare it with the laser hardened layer.
Fig.2.
The drawing is very indistinct and meaningless. The important processing point is minimized, while the large image of the laser head is shown - which does not contribute anything. Take a photo that reflects the treatment process - the visible object of the sample , machined plane, beam trajectory marked.
Fig. 3.
The microstructure is presented at higher magnification. The presented photo can only illustrate the dimensions of the blade in the cross-section, however, individual important fields on the cross-section should be marked and presented with larger enlargements.
Fig. 4.
The presented value of microhardness and its distribution, to my knowledge (I have carried out a lot of research on laser hardening of steel for thermal improvement), does not differ from traditional hardening in any way. With a given chemical composition, microhardness of approx. 1200 HV0.1 is obtained after laser hardening.
Fig. 5 and subsequent photos of microstructures.
It is difficult for me to comment on the photos, because in Fig. 3 there is no indication from which surface the distance was measured (from the tip line? Or from the side surfaces of the blade). It would have to be defined.
Other remarks:
On page 2, lines 68,69,70, the authors state that
"In this paper, 42CrMo press brake die was
quenched at 1133K and tempered at 553K, so the initial microstructure was tempered martensite. "This sentence indicates a misunderstanding of the physics of the laser hardening process.
The hardening temperature of 1133K (859.85 C) is the conventional hardening temperature of this steel grade, while laser heating in the micro-area results in much higher temperatures, and due to the large temperature gradient during cooling, a finer microstructure is obtained than in conventional hardening.
I have a question for the Authors: tempering after laser treatment The authors carried out at 553K (279.85 C) - what was the tempering process like and why at such a low temperature, because in the conventional treatment process, tempering is carried out in the temperature range of 540-680 C?
Author Response
Dear reviewers,
Thank you for your comments. Those comments are valuable and very helpful. We have read through comments carefully and have made corrections.
We would love to thank you for allowing us to resubmit a revised copy of the manuscript and we highly appreciate your time and consideration.
Sincerely.
Huizhen Wang
Response to Reviewer 2 Comments
Point 1: Table 1. Please indicate the standard according to which the chemical composition of the tested material was given, as well as specify the heat treatment after which the material was subjected to laser hardening.
Response 1: The chemical composition is based on GB/T 3077-2015, which was revised in this paper. Because the components marked in the text were originally quoted from the network, there was a little error in the components, so it was corrected. The composition of the material was tested before the laser surface hardening and confirmed as 42CrMo. The material was laser quenched without subsequent heat treatment. Please see the article for details.
Point 2: Fig. 1. The part subjected to laser treatment should be presented in accordance with the technical drawing - projections, dimensions. It is also necessary to mark on the drawing the laser processed surfaces, the trajectory of the laser beam, and the angle of incidence of the beam (it is known that the laser beam is directed perpendicular to the plane being processed.) In this case, when the edge of the element is rounded, it is quite a complicated machining procedure. If the orientation of the laser beam relative to the blade is perpendicular only to the tangent plane to the blade, you will never obtain a uniformly heated layer on the side surfaces of the blade and thus evenly hardened layers (positioning of the laser beam relative to the blade is shown in Fig. 3), as mentioned in article.
The figure should also show e.g. 1a. what the surface layer looks like after conventional hardening and tempering (its dimensions and microhardness), and then compare it with the laser hardened layer.
Response 2: Fig. 1 was drawn again, the relevant information of Fig. 2 and Fig. 3 was combined in Fig. 1, and Fig. 2 and Fig. 3 were deleted. The drawing the laser processed surfaces, the trajectory of the laser beam, and the angle of incidence of the beam were marked in figure 1. The figure 2b, figure 11 and figure 12 showed the uniformly microstructure and properties. In the future work, I hope to try to apply different direction of laser beam or other method to analyse the energy, microstructure and properties distribution. In the figure 1b, the initial hardness was added. The hardness after laser surface hardening is shown in figure 2. The size difference before and after laser surface quenching can not be detected because the specimen was cutting for the test of properties, but the surface quality did not change after laser quenching. I will pay attention to dimensional changes in subsequent work. Please see the fig.1 for details.
Point 3: Fig.2. The drawing is very indistinct and meaningless. The important processing point is minimized, while the large image of the laser head is shown - which does not contribute anything. Take a photo that reflects the treatment process - the visible object of the sample, machined plane, beam trajectory marked.
Response 3: The relevant information of Fig. 2 was combined in Fig. 1, and Fig. 2 was deleted. In fig. 1, the key dimensions and process parameters of sample are demonstrated. The fig.1 was drawing by CAD. Please see the fig.1 for details.
Point 4: Fig. 3.The microstructure is presented at higher magnification. The presented photo can only illustrate the dimensions of the blade in the cross-section, however, individual important fields on the cross-section should be marked and presented with larger enlargements.
Response 4: The relevant information of Fig. 3 was combined in Fig. 1, and Fig. 3 was deleted. The observation position coordinates were based on the coordinate system in Figure 1b and marked in the figures, and key information was added in fig. 1. Please see the fig.1 for details.
Point 5: Fig. 4.The presented value of microhardness and its distribution, to my knowledge (I have carried out a lot of research on laser hardening of steel for thermal improvement), does not differ from traditional hardening in any way. With a given chemical composition, microhardness of approx. 1200 HV0.1 is obtained after laser hardening.
Response 5: The advantages of laser quenching are high accuracy, high thermal concentration, low heat input, and high processing speed. In reference 1 and reference 2, 42CrMo were obtained respectively 710-750 HV0.1 and 660HV after laser hardening.
References
1 P. Sun, S. Li, G. Yu, et al. Laser surface hardening of 42CrMo cast steel for obtaining a wide and uniform hardened layer by shaped beams. Int J Adv Manuf Technol (2014) 70:787–796.
2 H. Su, B. Ma, Y. Yi, et al. Microstructure and properties of 42CrMo after laser surface melting and quenching. Ordnance Material Science and Engineering(2011)34:84-86.
Point 6: Fig. 5 and subsequent photos of microstructures.It is difficult for me to comment on the photos, because in Fig. 3 there is no indication from which surface the distance was measured (from the tip line? Or from the side surfaces of the blade). It would have to be defined.
Response 6: The observation position coordinates which were described as (X, Y) were based on the X-Y coordinate system in Figure 1b, marked in the figures, and further described in the figure title.
Point 7: On page 2, lines 68,69,70, the authors state that
"In this paper, 42CrMo press brake die was
quenched at 1133K and tempered at 553K, so the initial microstructure was tempered martensite. "This sentence indicates a misunderstanding of the physics of the laser hardening process.
The hardening temperature of 1133K (859.85 C) is the conventional hardening temperature of this steel grade, while laser heating in the micro-area results in much higher temperatures, and due to the large temperature gradient during cooling, a finer microstructure is obtained than in conventional hardening.
I have a question for the Authors: tempering after laser treatment The authors carried out at 553K (279.85 C) - what was the tempering process like and why at such a low temperature, because in the conventional treatment process, tempering is carried out in the temperature range of 540-680 C?
Response 7: The press brake die was quenched at 1133K and tempered at 553K. The hardness was 45HRC. Then, the blade was processed by laser surface hardening. The tempering temperature is determined according to the hardness requirements. The material supplier required the hardness to reach 47 ± 2HRC, so 553K was selected.

Reviewer 3 Report
Dear Authors,
Article is very interesting and its topic is suitable for “MDPI Coatings”. Paper structure is typical for scientific publications. Methodology and results are clearly presented. English is fine, however few minor mistakes (listed below) should be corrected.
Despite the quite good impression of the article the minor review should be performed:
- quality of each figure must be significantly improved, some of them (especially labels) are illegible, in pdf file its resolution is very low and it is unacceptable,
- Table 1 – page 2 – Was composition measured by Authors? Or it is from catalogue (if yes - please add the reference),
- line 60 (Section 2) – the standard number corresponding 42CrMo steel (ex. ISO, AMS, DIN ) should be cited,
- laser hardening process parameters are described in 2 sections (Materials and Methods , Results) by the same way (lines 70 – 71, 86 – 87). Authors should consider omit it in section results,
- figure 6 – please correct language mistake – it figure title should be Aspect (instead of Aspete) – correct it in y labels too,
- 141 – 144 – sentence – “In the misorientation distribution of Figure 9, it reflects that the frequency of 2°-5° grain boundaries in the hardening zone was also obviously higher than that in the substrate, which means the higher dislocation density existed in the hardening zone” – should be rewritten or deleted – this conclusion is obvious for readers.
- line 163 – replace show by shows,
- line 169 – replace word better by higher,
- process parameters (2mm, scanning speed of 1800mm/min and power of 2200W) are mentioned too many times (4 or 5) in paper (in each section besides introduction) which is unnecessary.
It must be clearly underlined that article is very interesting for potential readers and presents the high scientific level. After minor revision I will recommend it for publication in “Coatings”.
Best Regards
Author Response
Dear reviewers,
Thank you for your comments. Those comments are valuable and very helpful. We have read through comments carefully and have made corrections.
We would love to thank you for allowing us to resubmit a revised copy of the manuscript and we highly appreciate your time and consideration.
Sincerely.
Huizhen Wang
Response to Reviewer 3 Comments
Point 1: quality of each figure must be significantly improved, some of them (especially labels) are illegible, in pdf file its resolution is very low and it is unacceptable,
Response 1: I modified the pictures to make them clearer. Please see the article for details.
Point 2: Table 1 – page 2 – Was composition measured by Authors? Or it is from catalogue (if yes - please add the reference),
Response 2: The chemical composition is based on GB/T 3077-2015, which was revised in this paper. Because the components marked in the text were originally quoted from the network, there was a little error in the components, so it was corrected. The composition of the material was tested before the laser surface hardening and confirmed as 42CrMo. Please see the article for details.
Point 3: line 60 (Section 2) – the standard number corresponding 42CrMo steel (ex. ISO, AMS, DIN ) should be cited,
Response 3: The chemical composition is based on GB/T 3077-2015, which was revised in this paper. Please see the article for details.
Point 4: laser hardening process parameters are described in 2 sections (Materials and Methods , Results) by the same way (lines 70 – 71, 86 – 87). Authors should consider omit it in section results,
Response 4: The description of laser hardening process parameters of line 86 – 87 was deleted. Please see the article for details.
Point 5: figure 6 – please correct language mistake – it figure title should be Aspect (instead of Aspete) – correct it in y labels too,
Response 5: The mistake was revised. Please see the Fig.4 for details.
Point 6: 141 – 144 – sentence – “In the misorientation distribution of Figure 9, it reflects that the frequency of 2°-5° grain boundaries in the hardening zone was also obviously higher than that in the substrate, which means the higher dislocation density existed in the hardening zone” – should be rewritten or deleted – this conclusion is obvious for readers.
Response 6: Redundant description has been deleted. Please see the article for details.
Point 7: line 163 – replace show by shows,
Response 7: The mistake was revised. Please see the article for details.
Point 8: line 169 – replace word better by higher,
Response 8: The word was revised. Please see the article for details.
Point 9: process parameters (2mm, scanning speed of 1800mm/min and power of 2200W) are mentioned too many times (4 or 5) in paper (in each section besides introduction) which is unnecessary.
Response 9: Redundant description has been deleted. Please see the article for details.
